# Eruptive Styles Recognition Using High Temporal Resolution Geostationary Infrared Satellite Data

**Valerio Lombardo \*** , **Stefano Corradini, Massimo Musacchio** , **Malvina Silvestri** and **Jacopo Taddeucci**

Istituto Nazionale di Geofisica e Vulcanologia, 00143 Roma, Italy; stefano.corradini@ingv.it (S.C.); massimo.musacchio@ingv.it (M.M.); malvina.silvestri@ingv.it (M.S.); jacopo.taddeucci@ingv.it (J.T.)
* Correspondence: valerio.lombardo@ingv.it; Tel.: +39-06-51860-508

**Abstract:** The high temporal resolution of the Spinning Enhanced Visible and InfraRed Imager (SEVIRI) instrument aboard Meteosat Second Generation (MSG) provides the opportunity to investigate eruptive processes and discriminate different styles of volcanic activity. To this goal, a new detection method based on the wavelet transform of SEVIRI infrared data is proposed. A statistical analysis is performed on wavelet smoothed data derived from SEVIRI Mid-Infrared( MIR) radiances collected from 2011 to 2017 on Mt Etna (Italy) volcano. Time-series analysis of the kurtosis of the radiance distribution allows for reliable hot-spot detection and precise timing of the start and end of eruptive events. Combined kurtosis and gradient trends allow for discrimination of the different activity styles of the volcano, from effusive lava flow, through Strombolian explosions, to paroxysmal fountaining. The same data also allow for the prediction, at the onset of an eruption, of what will be its dominant eruptive style at later stages. The results obtained have been validated against ground-based and literature data.

**Keywords:** volcanic eruption interpretation; eruption forecasting; MSG SEVIRI; wavelet; remote sensing; thermal measurements; lava fountain; lava flow; Mt.Etna; eruptive style

## 1. Introduction

Determining the beginning and end of an eruption, and forecasting what will be the dominant eruption style, are equally important objectives, crucial for hazard assessment.

Satellite sensors have been increasingly employed for operational monitoring of volcanic thermal features as for example the geostationary Spinning Enhanced Visible and InfraRed Imager (SEVIRI) instrument on board the geostationary Meteosat Second Generation (MSG) satellite [1–4] and the Geostationary Operational Environmental Satellites (GOES) [5], or the polar Moderate Resolution Imaging Spectroradiometer (MODIS) [6], the Along Track Scanning Radiometer (ATSR) [7] and the advanced very-high-resolution radiometer (AVHRR) [3,8–12], on board respectively of the NASA Terra/Aqua, the ESA-ENVISAT, and the NOAA satellites.

In spite of its coarse spatial resolution, the high temporal rate of the geostationary systems represents an important tool for volcano monitoring (e.g., [13–15]). In particular, SEVIRI has 12 spectral channels from visible to thermal infrared, a spatial resolution of 3 km at nadir and a high temporal resolution that ranges from 15 min (earth full disk (EFD)) to 5 min (rapid scan mode (RSM) over Europe and Northern Africa). Exploiting these characteristics, SEVIRI allows a precise timing of the early phase of an eruption and almost a continuous monitoring of volcanic activity from the source to the atmosphere [4,16]. The SEVIRI data are collected in real time from a Meteosat-8 ground station antenna operating from 2010 at Istituto Nazionale di Geofisica e Vulcanologia (INGV) in Rome (Italy) [17,18].

In this work, the SEVIRI measurements have been used to investigate the eruptive processes and to discriminate different styles of volcanic activity by applying a novel procedure based on the correlation between the radiance growing rate at the beginning of the eruption with the specific type of volcanic activity. As test case, the 2013, 2015 and 2017 Etna events have been considered, and the results of the remote sensing analyses have been validated using ground-based information and literature references.

The paper is organized as follows: Section 2 outlines eruptions on Mt. Etna in the 2011–2015 period while Section 3 describes the method developed for the discrimination of the different styles of volcanic activities. In Sections 4 and 5 the results are presented and discussed, respectively. Final conclusions are drawn in Section 6.

## 2. The 2011–2015 and 2017 Etna Activities

In the 2011–2015 period, the eruptive style of Mt. Etna changed from predominantly effusive to more explosive activity. Strombolian events became more frequent and intense and were often associated with strong explosions, ash emissions, and lava fountaining paroxysmal episodes (e.g., [16]. The duration of these "paroxysmal" episodes varies from minutes to hours [4,19–22]. Between 2 and 8 December 2015, all four summit craters of Mt. Etna generated an extraordinary sequence of eruptive events. The activity consisted of high eruption columns, Strombolian explosions, lava flows, and widespread ash falls [23,24].

In February 2017, a vigorous strombolian activity started from the South-East (SE) crater producing a small lava flow. The direction of the lava flow was towards the southwest flank and flowed down slowly to an elevation of about 2850 m asl. After having lasted approximatively 48 h, the intense explosive-effusive activity began to weaken quickly. On 15 March new lava flows erupted from a fissure vent at the southern base of the SE crater complex. These lava flows descended into the *Valle del Bove* at several locations on the rim between 2500–2700 m elevation. On their descend down the steep western headwall, the lava flow fronts suffered partial collapses and interacted explosively with snow, both leading to the generation of small pyroclastic density currents that produced tall brown ash plumes [25].

## 3. Materials and Methodology

Statistical analysis was performed for Etnean eruptions using the 3.9 μm SEVIRI channel (Medium InfraRed-MIR channel) acquired from 2011 to 2017. Most of these eruptions were paroxysmal events, including Strombolian explosive activity, often associated with ash/gas emissions, and lava fountaining. Unfortunately, the statistics on lava flows are reduced to two episodes: the two-day long lava flow in December 2015 and the intense effusive activity that occurred from February to March 2017. The analysis of different eruptions has highlighted different radiance trends at the early stages of eruptions.

The December 2015 Etna eruption was characterized by the succession of several eruptive styles in a matter of days. Figure 1 shows the SEVIRI MIR radiance time series, measured from a single SEVIRI pixel centered over the Mt. Etna summit area and collected from 2 to 8 December 2015. The most powerful episode occurred between 2 and 3 December 2015 from the Voragine crater (Ep 1) when high lava fountains were produced without lava flows emitted. The MIR radiance drop that suddenly appeared in the MIR signal on 3 December, clearly indicates the absorption due to the ash emitted during the event [4]. From 4 to 5 December further powerful episodes occurred at the Voragine with strong emission of ash and gases (Ep 2). In this case, the scene was extremely cloudy and the radiances appear low compared to Ep 1 and its highly variable trend [4]. Finally, the activity shifted to the New Southeast Crater on 6 December 2015 (Ep 3), where Strombolian activity and lava flow emission lasted for two days and were fed by the most primitive magma of the study period, that is magma that underwent minimal igneous differentiation from the original composition [24]. Together with the MIR radiance trend, in Figure 1 are also drawn the growing rates for both the lava fountain (red dashed

curve) and the lava flow (blue dashed curve). The trend of the 2 and 3 December lava fountain shows a gradual rise to a peak in about 24 h with radiance growing at an exponential rate. The maximum intensity lasts less than one hour, after which the radiance quickly starts to decrease at an exponential rate. From 4 to 5 December, two further potent episodes occurred at the Voragine, characterized by ash emissions and weak Strombolian activity. Radiance trend shows a high variability with the presence of peaks in correspondence of paroxysmal episodes. However, the thick cloud coverage present on those days significantly influences the signal detected at the sensor. Therefore, it is not reliable to characterize this activity based on radiance changes [4]. The trend of the 6–8 December lava flow suddenly changes from steady to rising and the maximum radiance is reached after a few minutes from the beginning of the eruption. Then the radiance is maintained at the maximum value for the duration of the emissions.

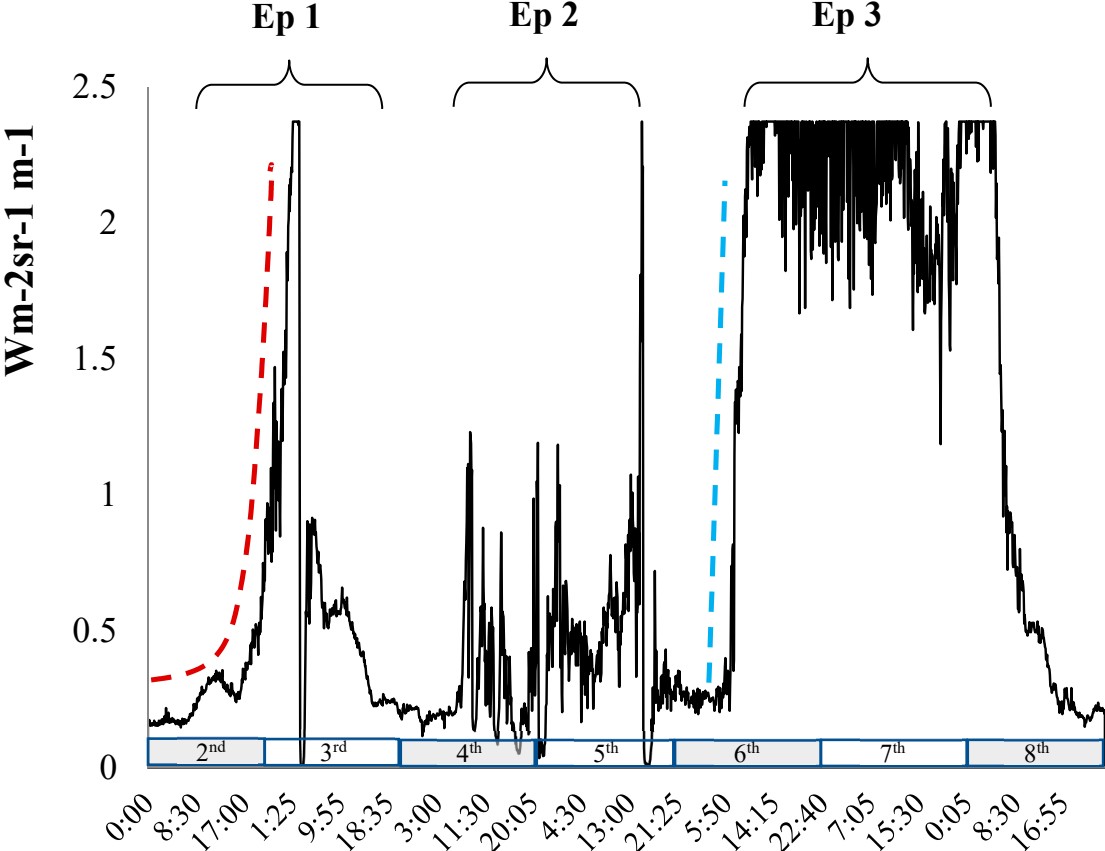

**Figure 1.** Spinning Enhanced Visible and InfraRed Imager (SEVIRI) Mid-Infrared (MIR) radiance acquired during the 2–8 December 2015 Mt. Etna eruption. Three different eruptive episodes occurred: lava fountaining (**Ep 1**), ash emission (**Ep 2**), and lava flow (**Ep 3**). Different episodes clearly show different trends that can be recognized a posteriori. However, differences are detectable even at the early stages of the eruption. An exponential trend and a sub-vertical trend seem to best fit the initial radiance increment for Ep1 (**red dashed line**) and Ep3 (**blue dashed line**) respectively.

The qualitative analysis of these three distinct episodes clearly shows strong differences in the radiance trends. It is quite easy to distinguish the three episodes knowing the overall trend retrospectively. However, for prevention reasons, it would be important to recognize the type of volcanic activity at the earliest stages of the eruption.

The basic idea is that when an eruption begins, the SEVIRI MIR radiance increases as a function of the temperature and the volume rate of the erupted products. There are two mechanisms that mainly contribute to defining the speed at which the integrated pixel temperature increases. On the one hand, the type of volcanic products generated by the eruption and their eruptive rates. For example, mixtures of ash, lava blebs and broiling gas at Mt. Etna are normally at temperatures between 200 °C and 700 °C,

but they can exceed temperatures of 1000 °C, while the measured eruptive temperatures of Etnean lava flows range between 1080 and 1095 °C [26]. On the other hand, products such as clouds, volcanic ashes, and plumes can act as a shield to radiation and partially or completely absorb the radiance emitted to space [27]. Both mechanisms affect the MIR radiance measured by the sensor, therefore, its trend has the potential to discriminate between eruptive styles. Herein, we aim to correlate the radiance growing rate at the beginning of the eruption (ramp) to the specific type of volcanic activity. By exploiting the SEVIRI RSM resolution a time-window of radiance values has been considered and analyzed in terms of signal processing. Changes in "signal" are detected at a certain time ($t_0$) using a moving window of fixed width, which is incremented at every SEVIRI acquisition.

Signal analysis is performed on SEVIRI MIR data in order to obtain optimal smoothing of radiance time-series. The smoothing removes noise introduced by radiometric calibration and atmospheric effects caused by passing of transitional clouds in the investigated scene. The literature on time-series smoothing and denoising techniques is vast and encompasses different methods. Among the most renown and utilized methods are the Savitzky-Golay and Hodrick-Prescott filters, the Hilbert-Huang transform (HHT), the ample class of kernel filters, as well as wavelet analysis. Herein, we propose a method that uses the wavelet transform [28] to analyze the SEVIRI time-series.

### 3.1. Wavelet Transform: the 'à trous' Algorithm

The *à trous* algorithm of discrete wavelet transform is a powerful tool for multiscale (multiresolution) analysis of images [28]. The *à trous* algorithm allows performing a hierarchical decomposition of an image into a series of *scale layers*, also known as *wavelet planes* [29]. Each layer contains only structures within a given range of characteristic *dimensional scales* in the space of a *scaling function*. The decomposition is done throughout a number of *detail layers* defined at growing characteristic scales, plus a final *residual layer* or *smoothing layer*, which contains the rest of unresolved structures. By isolating significant image structures within specific detail layers, detail enhancement can be carried out with high accuracy. Similarly, if noise occurs at some specific dimensional scales in the image, as is usual in most cases, by isolating it into appropriate detail layers we can reduce or remove it without affecting significant structures. A complete description of the implementation of this algorithm can be found in Bijaoui and Giudicelli [28]. Herein, the *à trous* algorithm has been adapted to remove the effect of background noise and to isolate changes due to volcanic activity in the moving time-window.

Plate 2a of Figure 2 shows a time-series of MIR radiance obtained for a short lava fountain event occurred at Mt. Etna from 13 to 15 December 2013. The detail layers and the residual layer from wavelet transform are shown in Plates 2b–e. Both detail and residual layers have potentials to be used for the assessment of eruptive phenomena. High-frequency changes occur at the earliest decomposition steps. Therefore, detail coefficients of the first scale layer are ideal to trigger the initial phase of the eruption (Plate 2b). One property of the wavelet transform is to have a sampling step proportional to the scale [28]. This implies that significant structures (e.g., local maxima) in lower-detailed scale layers match structures in higher-detailed scale layers (Plates 2b, c, d). This property allows "following" the eruptive signal in subsequent transformations where the higher frequency content of the signal is progressively eliminated.

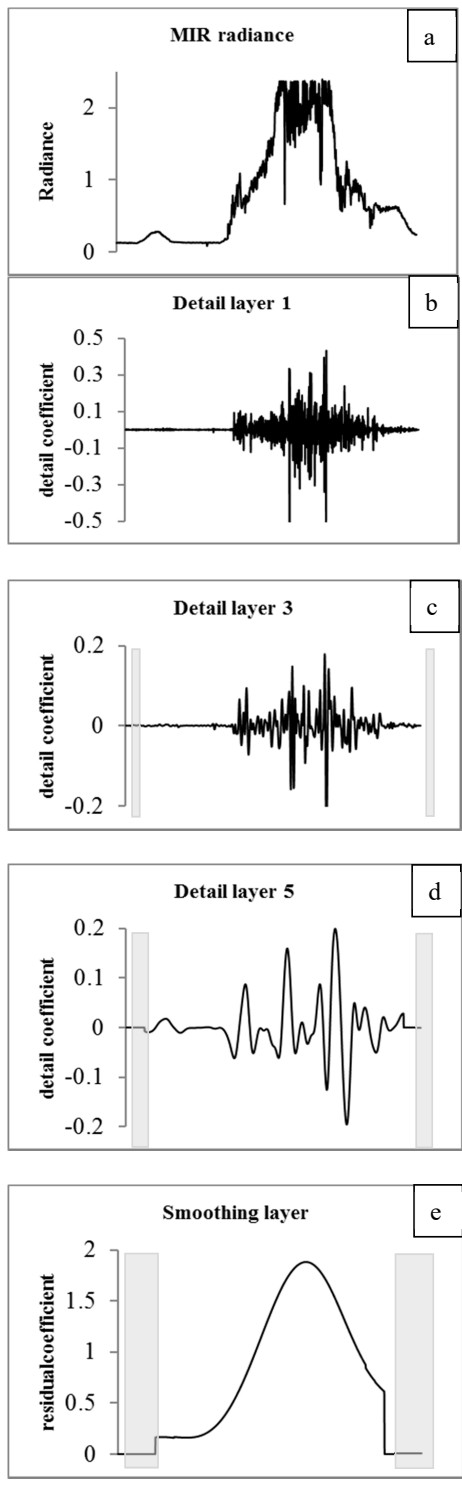

**Figure 2.** An example of multiscale decomposition with the *à trous* discrete wavelet transform algorithm. Seven detail layers have been generated with dyadic scaling sequence (only three are shown for clarity, **Plates 2b**, **2c**, and **2d**). Detail layers are generated for a growing scaling sequence of powers of two. The layers are generated for scales of 1, 2, 4, 8, . . . samples. For example, the fourth layer contains structures with characteristic scales between five and eight samples. Wavelet decomposition results in a border distortion which increases with the power of two at each decomposition scale (indicated in figures with grey areas). The original SEVIRI MIR radiance time-series is shown in **Plate 2a**. Detail layers at increasing characteristic scales contain larger signal structures. At the end of the sequence is the residual layer (**Plate 2e**), which contains all of the remaining unresolved structures.

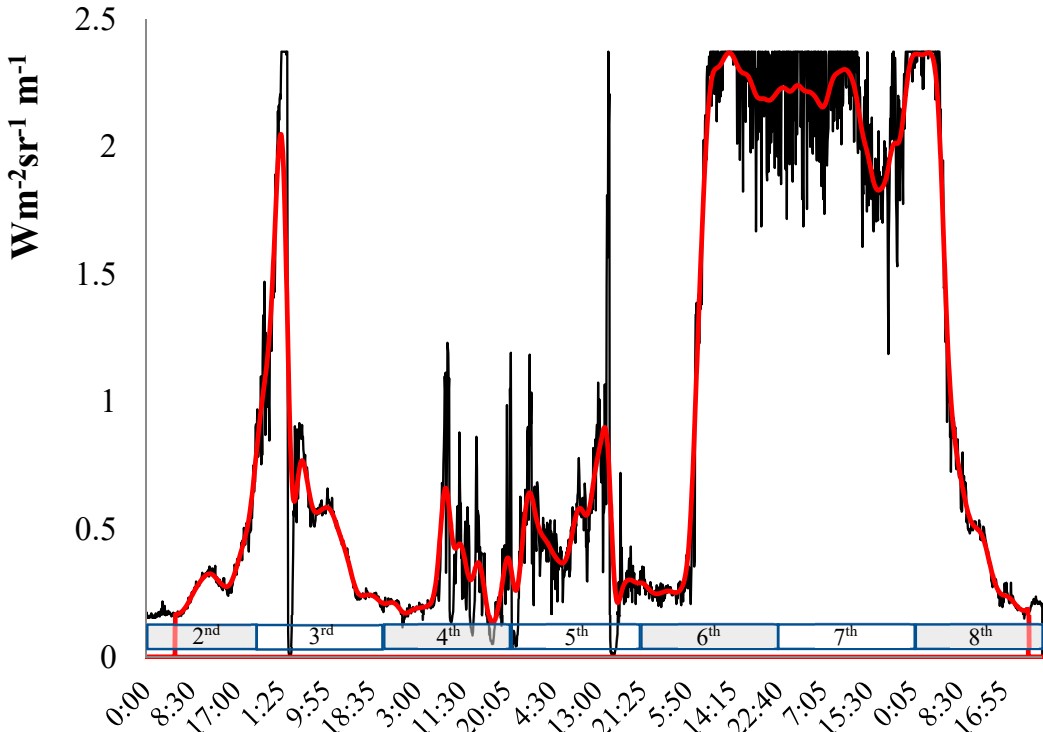

**Figure 3.** SEVIRI MIR radiance acquired during December 2015 Mt. Etna eruption. Red line is the smoothed layer derived from the *à trous* discrete wavelet transform. The smoothed layer allows for analysis of the overall trend without high-frequency noise due to instrumental noise, transient weather effects, and acquisition failures.

### 3.2. Statistical Analysis

The *smoothing layer* of the discrete wavelet transform allows capturing important patterns in the radiance data, while leaving out noise or other fine-scale structures/rapid phenomena. Herein, we propose a statistical approach to characterize the variability of the dataset in the *smoothing layer*.

Useful statistics for measuring the variability of a data set are skewness, kurtosis, and gradient. Skewness and kurtosis statistics can help us to assess certain kinds of deviations from normality of our data-generating process. Skewness is a measure of symmetry, or more precisely, the lack of symmetry. A distribution, or data set, is symmetric if it looks the same to the left and right of the center point. Skewness determines whether a distribution is symmetric about its maximum.

Kurtosis is defined as the degree to which a statistical frequency curve is peaked. Kurtosis is a measure of whether the data are heavy-tailed or light-tailed relative to a normal distribution. That is, data sets with high kurtosis tend to have heavy tails or outliers. Data sets with low kurtosis tend to have light tails or lack outliers. Finally, the gradient is defined as the slope of the line that interpolates the dataset of the moving window. The gradient measures the rate of change of radiance with time.

Time series analysis of SEVIRI data from 2011 to 2017 showed that kurtosis and gradient are most appropriate to characterize the relationship between radiance variations and eruptive styles.

## 4. Results

The sensitivities of the results were tested against different trends. Figure 4 shows the kurtosis (black line) and gradient (blue filled curve) values derived from *smoothing layer* (red line) at every SEVIRI acquisition (every 5 min) during the 2–8 December 2015 Etna eruption.

The kurtosis coefficient is often regarded as a measure of the tail heaviness of a distribution relative to that of the normal distribution. However, it also measures how much "peaked" a distribution is. When interpreting kurtosis, the normal distribution is used as a reference. A positive kurtosis

implies a distribution with more extreme possible data values (outliers) than a normal distribution, thus fatter tails (*Leptokurtic distributions*). A negative kurtosis implies a distribution with less extreme possible data values than a normal distribution, thus thinner tails (*Platykurtic distributions*). Finally, distributions with zero kurtosis have roughly the same outlier character as a normal distribution (*Mesokurtic distributions*).

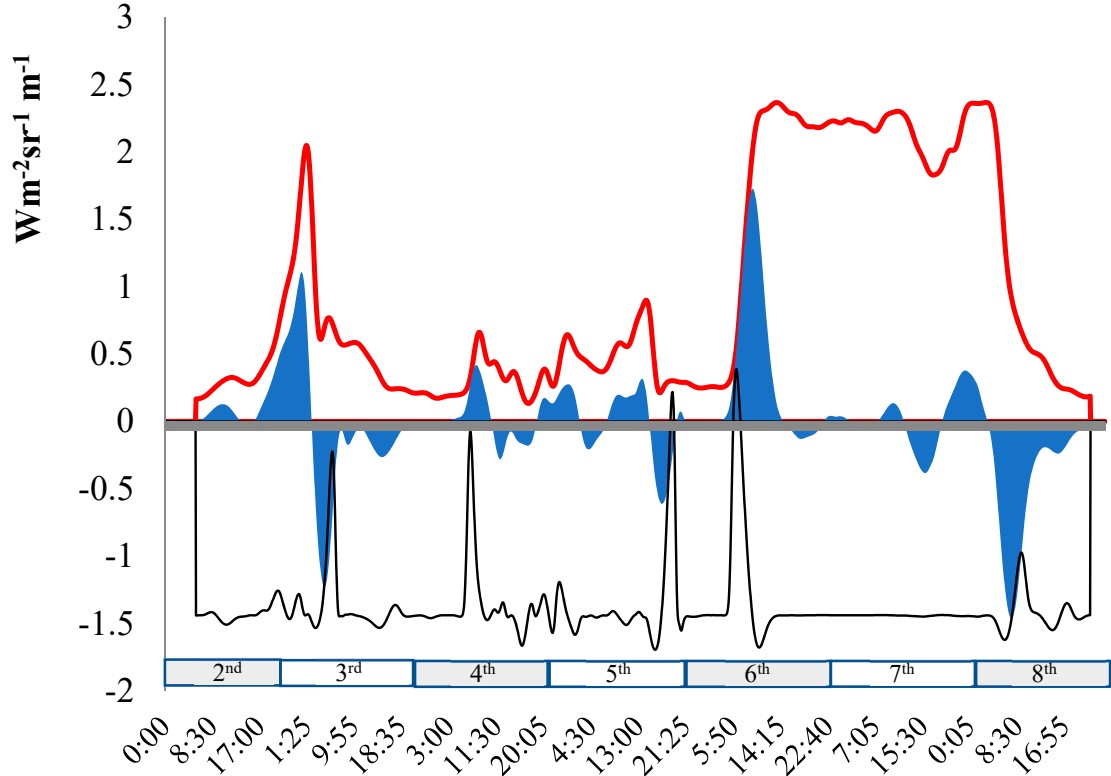

**Figure 4.** MIR radiance acquired during the December 2015 Etna eruption. The kurtosis (**black line**) and the gradient (**blue filled curve**) trends were derived from the *smoothing layer* (**red line**) at every Meteosat Second Generation (MSG) acquisition (every 5 min). Highest values of kurtosis (> 0) and gradient (> 1.5) occur at the beginning of the lava flow episode (6 December).

The ratio of kurtosis to its standard error can be used as a test of normality (that is, you can reject normality if the ratio is less than −2 or greater than +2). A large positive value for kurtosis indicates that the tails of the distribution are longer than those of a normal distribution. A negative value for kurtosis indicates shorter tails (becoming like those of a box-shaped uniform distribution).

Normalized kurtosis shows that the spread of the observations was slightly less than what would have been expected under the normal distribution. Kurtosis is less than −1.3 during non-eruptive phases and when the eruption reaches the highest point of intensity measurable with SEVIRI data. The normalized kurtosis trend indicates that outliers mainly occur at the beginning and the end of the eruptive episodes. Therefore, outliers in the kurtosis distribution can be used for hot-spot detection to identify the initial and final phases of the eruption. The magnitude of the outliers appears to be different depending on the eruptive episode. The magnitudes of the initial outliers are −1.4, −0.1, and 0.45 for the lava fountain, the explosive activity, and the lava flow episode, respectively.

The gradient (slope) of the line that best fits the *smoothing layer* distribution indicates the steepness of the *moving window* distribution at every SEVIRI acquisition. Positive slope indicates increments in radiance and, therefore, possible increases in volcanic activity. Again, the maximum slope at the onset of each eruptive episode appears to vary depending on the type of volcanic activity. A slope magnitude of 1.0, 0.4, and 1.7 is obtained at the onset peak for the three episodes.

Figure 5 shows the *smoothing layer* (red line), the normalized kurtosis (black line) and slope (blue filled curve) for the 28 February and the 15 March eruptive events. The two events, both characterized by the presence of lava flows, show a relatively high value of gradient and kurtosis at the onset of the eruption. The lava flow generated by the 28 February eruption displays a gradient of 2.1 and normalized kurtosis of 2.0. The lava flow of 15 March, much stronger and longer than the previous episode, shows maximum gradient and kurtosis values of 2.5 and 4.4, respectively.

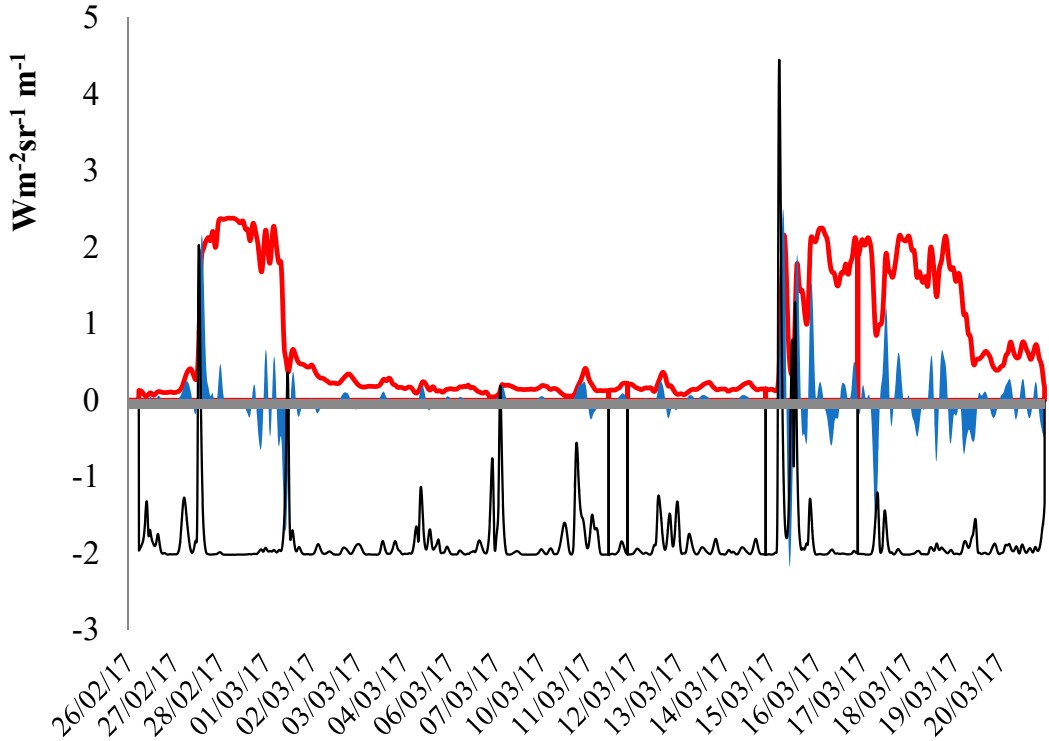

**Figure 5.** The *smoothing layer* (**red line**), the normalized kurtosis (**black line**) and slope (**blue filled curve**) for the 28 February and 15 March eruptive events. The lava flow generated by the 28 February eruption displays a gradient of 2.1 and normalized kurtosis of 2.0 at the onset of the eruption. The lava flow of 16 March, much stronger and longer than the previous episode, shows maximum gradient and kurtosis values of 2.5 and 4.4 respectively.

Threshold values were calculated using an eight-hour (100 samples) temporal window. The window width was experimentally chosen to optimize the signal-to-noise ratio of the radiance wavelet transform. A larger window implies that signal is smoothed over a longer period which results in losing details. A narrow window is more sensitive to high-frequency noise and, therefore, susceptible to false positives [30]. Figure 6 shows the *smoothing layer* (red line), the normalized kurtosis (black line) and slope (blue filled curve) for the 14 and 15 December 2013 Etna lava fountain. Vigorous Strombolian activity with associated ash plumes was observed at the New Southeast Crater (NSEC) (Volcano Discovery, Etna volcano activity updates: December 2013). The activity generated a small lava flow (probably mainly fed by rapid accumulation of liquid spatter) moving towards the SE flank with the presence of exploding magma bubbles towards the end of the eruption [31]. Under a statistical point of view, this paroxysmal event is very similar to the 3–5 December paroxysm (Ep 1) with kurtosis outlier of 0.05 and a maximum gradient of 0.45. Both episodes were characterized by Strombolian activity, intense ash emissions, and small effusive activity. Many of the paroxysms that took place during the investigation period fall into this type of scenario.

The transitional phases from Strombolian activity to lava fountaining can be observed and detected. On the evening of 16 March 2013, NSEC produced an intense episode of lava fountaining. This event, one of the most violent of the 2013 series of paroxysms, was preceded by a long "prelude"

of Strombolian activity [32] that started on the afternoon of 15 March and was followed by weak, discontinuous activity at the Voragine [31]. The transition from the strombolian to the lava fountaining episodes is recognizable in the slope (blue filled curve) trends of Figure 7. It is clear from the plots that we have an increase in the slope magnitude from 0.5 to 1.5 when the lava fountaining begins and the lava starts to overflow through the deep breach in the southeastern rim of the NSEC [33].

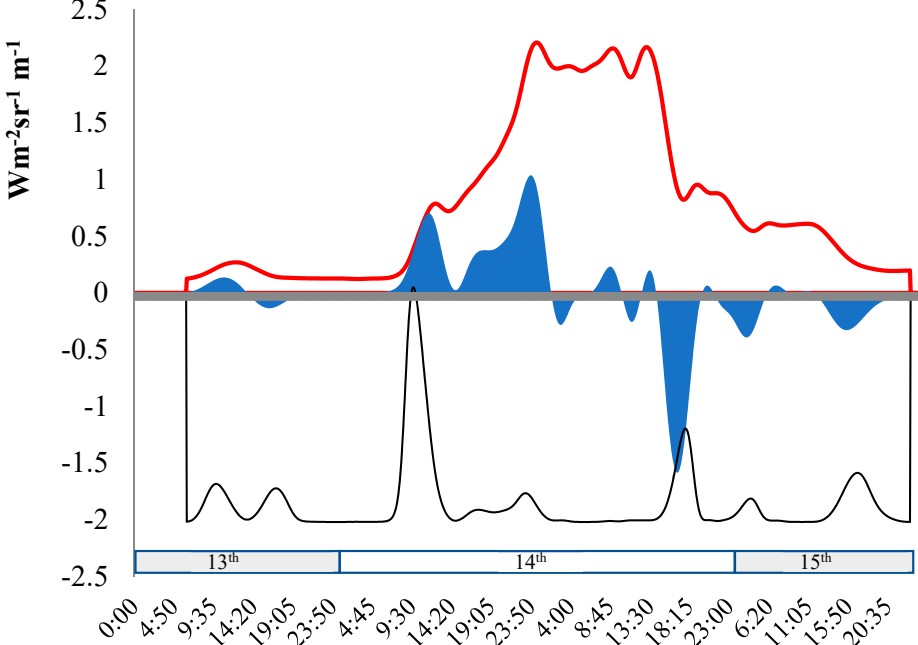

**Figure 6.** Statistics for the 14–15 December 2013 Etna lava fountain. Relatively low values of kurtosis (<0) and gradient (<0.5) characterize this kind of activity at the onset of the eruption.

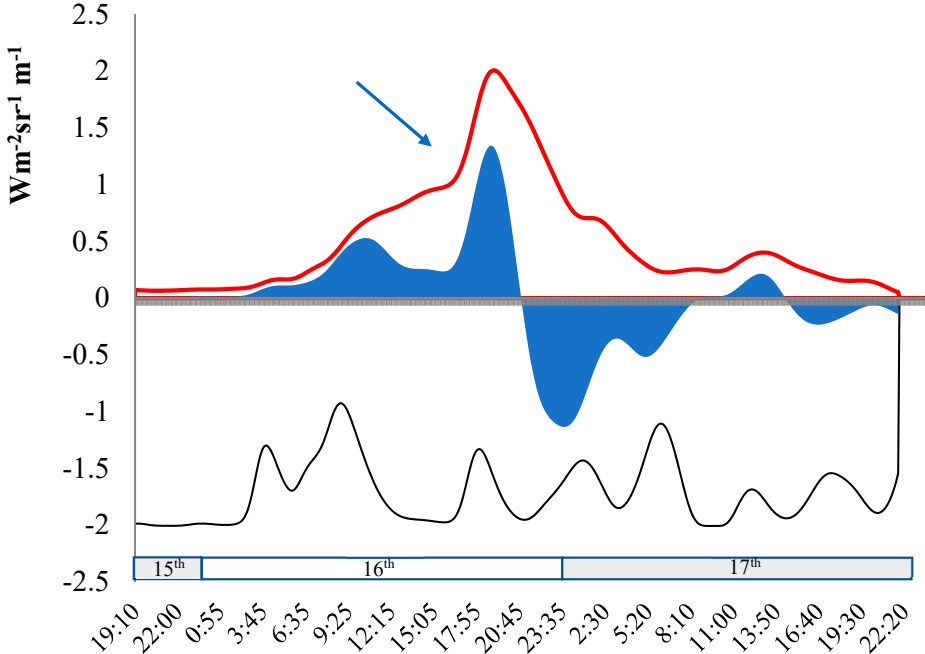

**Figure 7.** The analysis of statistical indexes allows highlighting transitions between different eruptive phases within the same activity. The transition from the Strombolian phase to the lava fountaining is well recognizable when we look at the peak (**blue arrow**) in gradient values (**blue filled curve**) during the 16 March 2013 Etna eruption.

## 5. Discussion

Our results suggest a relationship between the rate of increment in radiance (and thus temperature) and the nature of the volcanic process (i.e., eruptive styles as Strombolian activity, lava fountains and lava flow) causing that increment. Statistical analysis of SEVIRI MIR radiance trends highlighted the involvement of at least two different heating mechanisms for eruptions at Mt. Etna.

The first mechanism involves rapid increases in radiance and appears to be associated with effusive activity. Statistic kurtosis and gradient provide a measure of the steepness of the radiance trend. Relatively high values of kurtosis and gradient are mainly associated with the presence of active lava flows. In this case, Kurtosis and gradient magnitudes vary in the range of 0.0–5.0 and 0.5–3.0 respectively. We observed that statistics for small flows, as those generated by lava fountaining lava spatter or intra-crater lava ponding, show lower values of kurtosis (−0.5–0.5) and gradient (0.5–1) compared with values associated with long-term lava flows (see 2013 March case study of Figure 5).

The second mechanism is characterized by slope values usually less than 0.5 and refers to Strombolian events, often associated with paroxysmal activity and ash plumes. Moreover, the kurtosis outlier values remain below zero.

To understand and interpret the differences in radiance rates at the beginning of an eruption is not an easy task. However, we believe that such differences are the results of differences in eruptive and depositional processes of the different hot volcanic products (i.e., lava, ash, volcanic bombs). Eruption style controls both surface heating, the function of the output rate of lava and/or pyroclasts at a different temperature, and surface cooling, the function of the nature, temperature, and dispersal of the products. These heating and cooling mechanisms may be acting at the same time and thus explain the observed radiance behavior.

Temperatures of erupting magmas at Mount Etna are shown to be determined within a few °C in the range 1000–1200 °C [26]. The magmatic temperatures and petrological characters appear closely related to the volcanic activity. The alkaline basic magma of Mount Etna extrudes as lava at a nearly constant temperature of 1080 °C, but during explosive paroxysmal eruptions, the magma temperature is higher (1125 °C and possibly more). However, the small fragments of ash, lapilli, and bombs erupted during paroxysmal eruption fly through the air and cool quickly along their path through the atmosphere. Therefore, pyroclasts ejected in the atmosphere during explosive activity cool much faster than lava flows originated at effusive vents. In addition, gases and ash particles thrown into the atmosphere during volcanic eruptions influence the measured radiance. The particles spewed from volcanoes have the ability to absorb the radiation emitted by erupted materials shading the radiance signal detected at the sensor.

An important role is also played by the volume of the volcanic deposits generated during the eruption. The fraction of the pixel occupied by "hot products" depends on the type of volcanic eruptions and their spatial distribution. For example, explosive eruptions at Mt. Etna are mainly concentrated at the summit craters while lava flows tend to expand and occupy an increasingly larger area as the eruption continues. Therefore, the radiance detected at the sensor is a function of the temperature of the erupted materials and the fractional area occupied by those materials.

Validation was conducted by comparing our results with literature data using a time-series of SEVIRI data from 2011–2017 (4). We used threshold values of 0.1 and 0.5 for the kurtosis and the slope, respectively, to distinguish between effusive and explosive eruptions in the first 8, 16, and 24 h from the onset of the eruption. We found that, 79% of the eruptions were correctly interpreted in the first eight hours, 85% in the first 16 h, and 93% after a day from the beginning of the eruption.

However, several issues need to be taken into account. First, information on eruptions derived from monitoring bulletins is mainly qualitative information. Therefore, it is difficult to accurately determine the intensity of the eruptions and their duration from available literature data. Second, literature data are mainly referring to thermal camera data, remote sensing and field observations. These observations are all weather dependent. Third, our time-series suffers from data loss issues due to acquisition failure. Data loss and noisy images are suitable to generate artifacts that lead to

the presence of a false outlier. Finally, events occurring between 2011 and 2017 are mainly short-term paroxysmal eruptions, while long-term effusive eruptions are scarce.

The transition from Strombolian to fountaining activity is well documented [19,32]. Since 1999, Mount Etna has been characterized by episodic lava fountaining, each episode identified by initial Strombolian activity followed by transition to sustained fountaining to feed high effusion rate lava flow [32]. Such transitional behavior introduces further uncertainties into the identification and classification of the erupting style.

Mt. Etna, with its dense, multiparametric array of monitoring networks, provided an ideal test case for our analysis. Future integration of the proposed method will complement the existing networks and provide an additional monitoring tool independent from ground support. Finally, our results represent a promising step toward similar application to less monitored volcanoes, also in remote areas. Such application will require the threshold values empirically calculated for Mt. Etna to be adapted to specific cases or, alternatively, to be derived by ad-hoc models of eruption-dependent thermal radiation.

## 6. Conclusions

Eruption monitoring by satellite remote sensing still requires validation and refined, ground-tested methodologies. In this study, we explored the capability of high temporal resolution satellite data to discriminate and, to a limited extent, predict erupting styles at Etna volcano. A new statistical approach based on the wavelet transform of time-series of SEVIRI radiance data acquired at 5–15 min acquisition rate in the MIR spectral band (3.9 μm) is presented. Statistical analysis of processed data show potential in predicting eruptive styles. Statistic kurtosis and gradient derived from the analyzed time series allowed to recognize between two different eruptive styles: 1) paroxysmal events, characterized by Strombolian explosive activity, often associated with ash emissions, and lava fountaining (15–17/03/2013, 13–17/12/2013) and 2) lava flows (2–8/12/2015 and 26/02/2017–21/03/2017). It must be stressed that our methodology applies to data acquired at the onset of the eruption in order to give a quick response in terms of alert. The results obtained by this method have a considerable degree of trustworthiness in the first eight hours from the beginning of the eruption.

By considering the reliability of the proposed approach, the authors believe that it can be also applied worldwide, in particular for volcanoes lying in remote areas.

**Author Contributions:** V.L. conceived the idea, analyzed data, implemented wavelet transform, and wrote the manuscript with contributions from all authors. M.M. and M.S. processed satellite data and exported time-series for wavelet transform. S.C. and J.T. carried out the study of paroxysmal events with particular emphasis on the effects of gaseous volcanic emissions, ash plumes and explosive episodes. All authors contributed to the ideas, writing, and discussion.

**Funding:** This research received no external funding.

**Acknowledgments:** We thank the anonymous reviewers for their careful reading of our manuscript and their many insightful comments and suggestions.

**Conflicts of Interest:** The authors declare no conflict of interest.

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
