# Peer review of "Eruptive Styles Recognition Using High Temporal Resolution Geostationary Infrared Satellite Data"

_remotesensing, doi:10.3390/rs11060669_

Round 1

Reviewer 1 Report

The authors state that this technique can be used to differentiate three different types of activity at Etna. I question whether this adds to the monitoring efforts being done there. There are many other types of geophysical signals that can be used to differentiate eruptive style (from seismic, infrasound, radar, IR camera, etc.). They should state how this make a difference. I am skeptical that there is a universal kurtosis value that can be applied to remote volcanoes that lack monitoring instruments, and if so, how would that improve the near-real-time hazard assessment. 

The technique uses the radiance at one pixel over the summit of Etna. Could the authors estimate the percentage of the pixel that contains the active vent or lava flow? This is important as the fractional area of the pixel with lava is what determines the observed radiance. It could be that the kurtosis is due to the changes in the fractional area of very hot material, and not a change in eruption style.

When describing the December 2015 eruption, the authors refer to "primitive magma". This should be explained for the readers who are not volcanologists. Are you implying that this magma was hotter? If so, is it the hotter magma or the great fractional coverage of the one pixel that accounts for the radiance trends that are analyzed?

Not sure what to think about the radiance during the paroxysm.  Certainly the mid-IR radiance will be reduced due to the view being obscured by volcanic ash, but how that changes from eruption to eruption would be due to eruption rate (how opaque is the ash cloud), wind speed and direction (is is moving the ash plume away rapidly from the vent or is is stagnant. Were there meteorological clouds in the area during the eruption?

The authors use the terms fire and lava fountaining throughout the manuscript. This usage needs to be consistent. The correct term is lava fountaining.

The figures could be improved, especially the information shown on the X-axis. The labels are too crowded and repetitive. It would be very interesting to see the radiance values plotted against some other geophysical parameters like seismic and/or infrasound amplitude. Perhaps this kurtosis technique can be applied to multiple datasets to provide a more robust overall analysis of activity. 

Add time information the x-axis in Figure 2.

Figure 4 caption has "December: repeated.

Author Response

Reviewer 1

The authors state that this technique can be used to differentiate three different types of activity at Etna. I question whether this adds to the monitoring efforts being done there. There are many other types of geophysical signals that can be used to differentiate eruptive style (from seismic, infrasound, radar, IR camera, etc.). They should state how this make a difference. I am skeptical that there is a universal kurtosis value that can be applied to remote volcanoes that lack monitoring instruments, and if so, how would that improve the near-real-time hazard assessment. “

We partly agree with these observations. Etna is a natural scientific laboratory and it is one of the best studied volcanoes in the world. As such, it provided an ideal test case for our new methodology. Our results were compared with study of Ripepe et al., 2018 and Bombrun et al., 2016 on infrasound analysis at Mt. Etna showing a good agreement with their results. Currently, the monitoring network of Etna is comprised of hundreds of seismic, geodetic, and geochemical permanent sensors as well as thermal cameras and infrasound instrumentation. We believe that the addition of a monitoring tool independent of ground support is not redundant. Moreover, our technique aims to provide answers on those volcanoes that do not have broad geophysical networks and monitoring tools.

As a future work, we are planning a multidisciplinary approach to compare results obtained with our methodology with the results from other measurements such as seismology, geodesy and infrasound. We also plan to apply our methodology on different volcanoes.

We added to text:” Mt. Etna, with its dense, multiparametric array of monitoring networks, provided an ideal test case for our analysis. Future integration of the proposed method will complement the existing networks and provide an additional monitoring tool independent from ground support. Finally, our results represent a promising step toward similar application to less monitored volcanoes, also in remote areas. Such application will require the threshold values empirically calculated for Mt. Etna to be adapted to specific cases or, alternatively, to be derived by ad-hoc models of eruption-dependent thermal radiation.“

The technique uses the radiance at one pixel over the summit of Etna. Could the authors estimate the percentage of the pixel that contains the active vent or lava flow? This is important as the fractional area of the pixel with lava is what determines the observed radiance. It could be that the kurtosis is due to the changes in the fractional area of very hot material, and not a change in eruption style.

This is a very interesting point!

Sub-pixel temperature techniques are usually applied on low-spatial resolution pixel to retrieve different thermal components within the pixel. In the case of pixel occupied by lava we want to calculate at least the lava temperature, the background temperature and the fractional area occupied by lava. The number of thermal components depends on the number of distinct bands and many other factors (Oppenheimer 1997, Harris et al. 2007, Lombardo et al, 2016, 2014; Wright et al., 2011, and many others).

To solve the so called “Dual-band system” we need at least two IR bands. However, there are many issues when applying the dual-band technique on SEVIRI data:

1)     We have no anomaly in SEVIRI TIR band. Therefore, we can use MIR band only.

2)     following Wright and Flynn (2004), we can use the MIR radiance equivalent for the TIR pixel-integrated temperature to approximate the radiance of the ambient temperature for surfaces within the MIR pixel. However, this introduce a big approximation.

3)     The lava fractional area may be calculated following the equation from Harris et al. 2007. Again, we obtain large errors (40%) on estimation of the fraction. When we apply this methodology to SEVIRI data we observe that lava fractional area tends to increase or decrease every 5 minutes!

Therefore, we decided to analyze the (smoothed) temporal trend rather than trying to retrieve information from a single MSG acquisition.

We believe that increments in the lava fractional area play an important role in recognizing eruptive style as stated in the test:

“An important role is also played by the volume of the volcanic deposits generated during the eruption. The fraction of the pixel occupied by “hot products” depends on the type of volcanic eruptions and their spatial distribution. For example, explosive eruptions at Mt. Etna are mainly concentrated at the summit craters while lava flows tend to expand and occupy an increasingly larger area as the eruption continues. Therefore, the radiance detected at the sensor is a function of the temperature of the erupted materials and the fractional area occupied by those materials.”

However, due to the characteristics of SEVIRI data, it is pretty difficult to derive precise information on the fractional lava from MIR radiance trends.

When describing the December 2015 eruption, the authors refer to "primitive magma". This should be explained for the readers who are not volcanologists. Are you implying that this magma was hotter? If so, is it the hotter magma or the great fractional coverage of the one pixel that accounts for the radiance trends that are analyzed?

We added to text: “..that is magma that underwent minimal igneous differentiation from original the composition”

Although the most primitive magma is less differentiated than the previously erupted basalt, the average eruptive temperature for Etna tends to remain between 1080 - 1095°C (Pompilio,  1998).

Not sure what to think about the radiance during the paroxysm.  Certainly the mid-IR radiance will be reduced due to the view being obscured by volcanic ash, but how that changes from eruption to eruption would be due to eruption rate (how opaque is the ash cloud), wind speed and direction (is is moving the ash plume away rapidly from the vent or is is stagnant. Were there meteorological clouds in the area during the eruption?

Our approach is a statistical approach. We have no way of removing transient atmospheric effects from our data. For this reason, we have used wavelets to filter data and to obtain trends as representative as possible of the volcanic phenomenon.  

The authors use the terms fire and lava fountaining throughout the manuscript. This usage needs to be consistent. The correct term is lava fountaining.

Lava fountaining is now used throughout the manuscript.

The figures could be improved, especially the information shown on the X-axis. The labels are too crowded and repetitive. It would be very interesting to see the radiance values plotted against some other geophysical parameters like seismic and/or infrasound amplitude. Perhaps this kurtosis technique can be applied to multiple datasets to provide a more robust overall analysis of activity. 

The X-axis now shows the time and dates are visualized in different color on the images for clarity purposes.

Reviewer 2 Report

This paper introduces a new method to detect the onset of volcanic activity and to classify between different eruptive style using mid-infrared satellite observations. The application of the method is demonstrated for a number of eruptions of Mount Etna, Italy in 2011 to 2017, using SEVIRI satellite observations.

Overall, I found this paper interesting to read and I think that it fits well in the scope of Remote Sensing. I would recommend it for publication subject to a number of comments and corrections.

As the manuscript does not have page numbers and line numbers, I've put my comments directly in the attached pdf file.

Author Response

Reviewer 2

All text issues have been addressed.

At this point, I was wondering if you are looking at cloud-free/filtered measurements, only “

Our approach is a statistical approach. We have no way of removing transient atmospheric effects from our data. For this reason, we have used wavelets to filter data and to obtain trends as representative as possible of the volcanic phenomenon.

“You are analysing radiances, but it wouldn't it be more interesting to look at brightness temperatures, because they are a more direct measure of temperature”

This makes sense. However, the temperature measured from a 5 km SEVIRI pixel would be an average of different thermal components such as lava (crust + molten fraction), ambient temperature, clouds, and volcanic plumes and gas emissions. Therefore, we think that the temperature so reported might be misinterpreted.

“Why not simply use a radiance threshold to detect volcanic activity”

When an eruption begins, the kurtosis values become very “peaky”. Therefore, kurtosis provide a key to precisely detect the initial phase of eruptions. Kurtosis is also a signed parameter that allow you to identify the beginning and the end of the eruption. This is useful when implemented in an automatic detection routine.

Did other studies use kurtosis + gradient for time series analysis? Would your method work well for other cases, other volcanoes?

No, not yet. But we plan to apply our methodology on other volcanoes as a future work.